# Risk Factors for Therapeutic Intervention of Remdesivir in Mild to Moderate COVID-19—A Single-Center Retrospective Study of the COVID-19 Fourth Pandemic Period in Wakayama, Japan

**DOI:** 10.3390/medicina58010118

**Published:** 2022-01-13

**Authors:** Shinobu Tamura, Takahiro Kaki, Mayako Niwa, Yukiko Yamano, Shintaro Kawai, Yusuke Yamashita, Harumi Tanaka, Yoshinobu Saito, Yoshinori Kajimoto, Yusuke Koizumi, Hiroki Yamaue, Naoyuki Nakao, Takako Nojiri, Masaya Hironishi

**Affiliations:** 1Department of Internal Medicine, Wakayama Medical University Kihoku Hospital, Wakayama 6497113, Japan; kakitaka@wakayama-med.ac.jp (T.K.); m-niwa@wakayama-med.ac.jp (M.N.); yukiko1106.y@gmail.com (Y.Y.); skawai@wakayama-med.ac.jp (S.K.); y-kajimo@wakayama-med.ac.jp (Y.K.); msyhrns@wakayama-med.ac.jp (M.H.); 2Department of Hematology/Oncology, Wakayama Medical University, Wakayama 6418509, Japan; yyyamash@wakayama-med.ac.jp; 3Department of Internal Medicine III, Wakayama Medical University, Wakayama 6418509, Japan; 4Department of Neurology, Wakayama Medical University, Wakayama 6418509, Japan; 5Department of Nephrology, Wakayama Medical University, Wakayama 6418509, Japan; 6Department of Nursing, Wakayama Medical University Kihoku Hospital, Wakayama 6497113, Japan; harumi-t@wakayama-med.ac.jp; 7Department of Pharmacy, Wakayama Medical University Kihoku Hospital, Wakayama 6497113, Japan; saitoh@wakayama-med.ac.jp; 8Department of Infection Control and Prevention, Wakayama Medical University, Wakayama 6418509, Japan; ykoizumi@wakayama-med.ac.jp; 9Second Department of Surgery, Wakayama Medical University, Wakayama 6418509, Japan; yamaue-h@wakayama-med.ac.jp; 10Department of Neurosurgery, Wakayama Medical University, Wakayama 6418509, Japan; nnakao@wakayama-med.ac.jp; 11Wakayama Prefecture, Health Department, Wakayama 6408585, Japan; nojiri_t0003@pref.wakayama.lg.jp

**Keywords:** COVID-19, B.1.1.7 variant, hospitalization, remdesivir, headache

## Abstract

*Background and Objectives:* The incidence of coronavirus disease 2019 (COVID-19) has increased in Wakayama, Japan, due to the spread of the highly infectious B.1.1.7 variant. Before this event, the medical systems were almost unaffected. We aimed to assess the clinical characteristics of patients hospitalized with COVID-19 and the risk factors for therapeutic intervention of remdesivir during the fourth pandemic period in Wakayama, Japan. *Materials and Methods:* This single-center retrospective study enrolled 185 patients with mild to moderate COVID-19 hospitalized in our hospital without intensive care between 14 March and 31 May 2021. *Results:* In this period, 125 (67.6%) of the 185 patients had the B.1.1.7 variant. Sixty-three patients (34.1%) required remdesivir treatment. Age upon admission and length of hospitalization were significantly different between remdesivir treatment and careful observation groups (mean (standard deviation); 59.6 (14.7) versus 45.3 (20.6) years; *p* < 0.001 and median (interquartile range); 10 (9–12) versus 9 (8–10) years; *p* < 0.001). One patient was transferred to another hospital because of disease progression. At hospital admission, age ≥60 years (odds ratio (OR) 6.90, *p* < 0.001), a previous history of diabetes mellitus (OR 20.9, *p* = 0.002), B.1.1.7 variant (OR 5.30; *p* = 0.005), lower respiratory symptoms (OR 3.13, *p* = 0.011), headache (OR 3.82, *p* = 0.011), and fever ≥37.5 °C (OR 4.55, *p* = 0.001) were independent risk factors to require remdesivir treatment during the admission. Conclusions: Many patients with mild to moderate COVID-19 required the therapeutic intervention of remdesivir during the fourth pandemic period in Wakayama, Japan. From the clinical data obtained at admission, these risk factors could contribute to a prediction regarding the requirement of remdesivir treatment in cases of mild to moderate COVID-19.

## 1. Introduction

The coronavirus disease 2019 (COVID-19) infection broke out at the end of 2019 and has since spread globally. The World Health Organization (WHO) declared a pandemic on 11 March 2020 [1]. In Japan, the number of patients with COVID-19 also increased, and on 7 April 2020, the Government declared a state of emergency to prevent further infections. The two emergency declarations by the Government were temporarily effective. We experienced three pandemic periods until February 2021 [2]. The fourth pandemic epi-curve, starting in March 2021, was even larger due to the prevalence of the highly infectious variant B.1.1.7 strain [3]. As of May 2021, remdesivir (an antiviral agent), dexamethasone, and baricitinib (anti-inflammatory agents) were available in Japan. Despite these approved drugs, patients with the mild illness often experienced disease progression or death [4].

The public health and medical administration of the Wakayama Prefecture followed the basics of the Japanese Infectious Diseases Control Law, i.e., early detection, early isolation, and a thorough investigation of behavioral history. Our method, the “Wakayama model”, aimed at an early and thorough investigation of the cluster tree and complete hospitalization of all infected cases regardless of symptoms. With the help of several institutions of different functions, we accumulated more than 400 beds, which was adequate capacity for the entire population in this prefecture. The Wakayama Medical University Kihoku Hospital (referred to as Kihoku Hospital) is a designated hospital for patients with mild to moderate COVID-19 symptoms in the Wakayama Prefecture.

The baseline characteristics and optimal timing to begin the treatment intervention for patients with mild to moderate COVID-19 symptoms in daily clinical practice remain unclear. In the current study, we described the clinical characteristics and outcomes of patients hospitalized with COVID-19 during the fourth pandemic period in Wakayama, Japan. Furthermore, we aimed at exploring the risk factors requiring remdesivir treatment in these hospitalized patients with mild to moderate COVID-19 infection.

## 2. Materials and Methods

### 2.1. Study Participants

During the COVID-19 pandemic, the Wakayama Prefecture administration provided hospital protocols for admitting all patients with COVID-19 living in the area regardless of disease severity. For all patients, COVID-19 diagnosis was confirmed using polymerase chain reaction (PCR) for severe acute respiratory syndrome coronavirus 2 (COVID-2) via nasopharyngeal swabbing. Based on the WHO Regional Office for South-East Asia COVID-19 Health System Response Monitor, patients with COVID-19 were categorized into four grades [4]. Cases with no respiratory symptoms and oxygen saturation (SpO_2_) ≥ 96% were considered mild. Cases with pneumonia and SpO_2_ between 93% and 96% were graded as Moderate I, whereas the requirement for supplemental oxygen (SpO_2_ ≤ 93% on room air) was graded as Moderate II. Cases requiring intensive care were regarded as severe.

The Wakayama Red Cross Hospital, Wakayama Rosai Hospital, and Wakayama Medical University Hospital preferentially accepted patients with severe COVID-19. Since Kihoku Hospital has no intensive care units, we mainly accepted patients with mild to Moderate I COVID-19 based on the Wakayama Prefecture administration recommendations. In total, 185 patients (residents of Wakayama) were admitted to Kihoku Hospital and were discharged between 14 March 2021, and 31 May 2021. This study enrolled COVID-19 cases admitted to our institution, and a single-center retrospective analysis was performed according to treatment interventions. No enrolled patients had taken routine prophylactic administration of oral antipyretic or analgesic medications. This study was conducted by the tenets of the Declaration of Helsinki and was approved by the institutional review board of Wakayama Medical University.

### 2.2. Clinical Data

Data regarding the characteristics of patients with COVID-19 were extracted from medical records and were then reviewed retrospectively. We focused on sex, age, comorbidities, smoking status, body mass index (BMI), COVID-19 genotypes, various symptoms, and vital signs. The comorbidities included hypertension (HT), diabetes mellitus (DM), hyperlipidemia (HL), chronic obstructive pulmonary disease (COPD), chronic kidney disease (CKD), heart disease, and cancer. No patient had previous primary headaches. The presence of smoking history comprised both smokers and ex-smokers. Common symptoms associated with COVID-19 were upper respiratory symptoms (such as nasal discharge and throat pain), lower respiratory symptoms (such as cough and sputum), general fatigue, headache, arthralgia, olfactory, and taste disturbances, gastrointestinal symptoms, and shortness of breath. These symptoms at admission were obtained from the questionnaire (Appendix A). The initial vital signs consisted of temperature, SpO_2_, and pulse rate.

### 2.3. Treatment Strategy in Kihoku Hospital

In our hospital, the COVID-19 treatment strategy was based on the official Japanese guideline developed by the Japanese Ministry of Health, Labor, and Welfare. Briefly, patients with mild COVID-19 were hospitalized for careful monitoring. Remdesivir treatment was performed for patients with Moderate I to II COVID-19 diagnoses. Severe patients were transferred to other hospitals with intensive care units. The patients underwent early diagnosis and treatment. All patients received 200 mg intravenous remdesivir on Day 1, followed by 100 mg daily on Days 2–5 [5,6]. Most patients who had or had experienced progression to Moderate I COVID-19 received this treatment. Further, a few mild cases without pneumonia and desaturation received this treatment since these patients had severe nausea, diarrhea, and general fatigue. Soon after the use of baricitinib was approved in Japan on 23 April 2021, patients who required supplemental oxygen were also treated with 4 mg oral baricitinib [6,7]. Some of these patients with residual and pulmonary symptoms received 4 mg baricitinib once daily for several days.

### 2.4. Statistical Analysis

All statistical analyses were performed using EZR software (Eazy R) version 1.55, which is based on R and R commander (Saitama Medical Center, Jichi Medical University, Saitama, Japan) [8]. Continuous variables were reported as the means and standard deviations (SD) or medians and interquartile range (IQR) if they were non-normally distributed, as evaluated by the Shapiro–Wilk normality test. Categorical variables were presented as numbers with percentages. Differences between continuous variables were evaluated using a *t*-test or the Wilcoxon rank-sum test, and differences between the proportions of categorical variables were assessed using the Chi-squared test or the Fisher’s exact test. Univariate and multivariate logistic regression analyses were conducted to examine the relationships between treatment intervention and multiple independent factors. For all statistical analyses, a *p*-value of <0.05 was considered significant.

## 3. Results

### 3.1. Characteristics of Treated Patients with COVID-19 during the Fourth Pandemic Period

During the fourth pandemic period (from 14 March to 31 May 2021), 185 patients were admitted to our institution. Table 1 shows the clinical characteristics of these patients. Before the fourth pandemic period, 17 (9.6%) of 177 patients were treated with remdesivir. During the fourth pandemic period, 122 (65.9%) of 185 patients were carefully observed without treatment (also called the observation group), whereas 63 (34.1%) required the treatment intervention (also called the treatment group). More than half of these patients were men (57.1%). The mean age of hospitalized patients was 49.3 years (SD, 19.8). Moreover, 81 (43.8%) patients had comorbidities. Common comorbidities were HT (24.9%, 24 cases), followed by DM (9.2%, 17 cases). Of 185 patients examined in the fourth pandemic period, 125 (67.6%) were infected with the B.1.1.7 variant. The variant was detected in 49 (77.8%) of 63 treated patients. In this pandemic period, lower respiratory symptoms (34.1%; 63 cases) were the most common symptom, followed by upper respiratory symptoms (29.2%; 54 cases), general fatigue (17.3%; 32 cases), and headache (15.7%; 29 cases). Among 63 hospitalized patients requiring the treatment intervention, the mean temperature and the median SpO_2_ at admission were 38.0 °C (SD, 0.84) and 95% (IQR, 94–95). Moreover, the initial mean pulse of these patients was 87/min (SD, 13/min).

All patients upon treatment initiation underwent basic laboratory tests. The laboratory findings of most patients improved after five days of treatment. Of the 63 patients treated with remdesivir, three experienced mild adverse effects (*n* = 2, Grade 1 hepatotoxicity and *n* = 1, Grade 1 nausea according to CTCAE version 5.0). In total, 12 (19.0%) patients with Moderate II COVID-19 were treated with baricitinib. Fifty-seven patients (90.5%) were treated with heparin calcium via subcutaneous injection every 12 h.

Of 63 patients treated, 40 (63.5%), 19 (30.2%), and four (6.3%) presented with mild, Moderate I, and Moderate II COVID-19 upon admission, and five (7.9%), 41 (65.1%), and 17 (27.0%) upon treatment initiation, respectively.

### 3.2. Comparisons of Clinical Characteristics between the Treatment Requirement and the Observation Groups in Our Hospital

Table 2 compares the treatment requirement and the observation groups according to 23 clinical categories (sex, age, comorbidities, smoking status, BMI, COVID-19 genotypes, symptoms, and vital signs) at admission. These two groups did not significantly differ in terms of sex, smoking status, and BMI (≥30 kg/m^2^). As a result, the proportion of patients aged ≥60 years and those with four comorbidities (HT, DM, COPD, and cancer) and infected with the B.1.1.7 variant were significantly higher in the treatment-requirement group than in the observation group (*p* < 0.05). Among the symptoms associated with COVID-19, lower respiratory symptoms, general fatigue, headache, gastrointestinal symptoms, and shortness of breath were also significantly higher in the treatment-requirement group (*p* < 0.05). Moreover, more patients requiring the treatment intervention had a fever ≥ 37.5 °C, SpO_2_ < 97%, and pulse ≥ 90/min.

### 3.3. Clinical Outcome of Hospitalized Patients with COVID-19

Next, we compared the treatment requirement and the observation groups focusing on the outcome (Table 1). In this analysis, 21 patients during the early fourth pandemic period were excluded due to differences in discharge criteria. There was a significant difference in terms of age upon admission between the treatment-requirement and the observation groups (mean (SD); 59.6 (14.7) versus 45.3 (20.6) years; *p* < 0.001). The median duration from disease onset to admission did not differ between the two groups: 2 days (IQR, 2–4) in the treatment-requirement group vs. 2 days (IQR, 1–4) in the observation group (*p* = 0.12) (Table 1). The treatment-requirement group was hospitalized slightly longer than the observation group (median (IQR), 10 days (9–12) vs. 9 days (8–10); *p* < 0.001) (Table 1). In the treatment-requirement group, the median duration (IQR) from onset to treatment initiation was 6 days (4–7). In this period, one treated patient was transferred to another hospital with an intensive care unit. None of the patients died in this study.

### 3.4. Factors Associated with the Therapeutic Intervention of Remdesivir in Hospitalized Patients with Mild to Moderate COVID-19

As shown in Table 2, 23 clinical categories were recorded in these patients with mild to moderate COVID-19 at admission. Among these categories, univariate logistic regression analysis was carried out to identify the risk factors for therapeutic intervention of remdesivir. Table 3 depicts the results of the univariate analysis. Older age, two comorbidities (hypertension and diabetes mellitus), smoking history, the B.1.1.7 variant, four symptoms at admission (lower respiratory symptoms, general fatigue, headache, and gastrointestinal symptoms), and initial vital signs (fever ≥ 37.5 °C, SpO_2_ < 97%, pulse ≥ 90/min) were extracted as significant factors for the treatment-requirement group (*p* < 0.05).

Next, most variables that were significant in the univariate logistic regression (*p* < 0.001) were chosen as candidates for multivariate logistic regression analysis. These variables were age, two comorbidities, the B.1.1.7 variant, four symptoms, and temperature at admission. In the multivariate analysis with adjusted odds ratio (OR), ≥ 60 years of age (OR: 6.90; 95% confidence interval [CI]: 2.57–18.0; *p* < 0.001), diabetes mellitus (OR: 20.9; 95% CI: 3.11–140; *p* = 0.002), the B.1.1.7 variant (OR: 5.30; 95% CI: 1.65–17.0; *p* = 0.005), lower respiratory symptoms (OR: 3.13; 95% CI: 1.30–7.53; *p* = 0.011), headache (OR: 3.82; 95% CI: 1.37–10.6; *p* = 0.011), and fever ≥ 37.5 °C (OR: 4.55; 95% CI: 1.83–11.3; *p* = 0.001) were found to be independent risk factors for requiring remdesivir treatment among hospitalized patients with mild to moderate COVID-19 (Table 4, Appendix A).

## 4. Discussion

The Japanese guidelines recommend the use of remdesivir among hospitalized patients with moderate to severe illness [4]. In the fourth pandemic period, in our hospital, 63 patients were treated with remdesivir. In terms of clinical characteristics, most patients were aged ≥60 years and had underlying comorbidities including HT and DM. Furthermore, COVID-19 infection was characterized by systemic symptoms including general fatigue, headache, and gastrointestinal symptoms as well as lower respiratory symptoms. These findings are similar to those reported in previous studies [9,10,11]. In moderate COVID-19, the median duration of symptoms from onset to treatment initiation was reported to be eight days among patients receiving a 5-day course of remdesivir [12]. Moreover, Mehta et al. reported that in patients with moderate to severe illness, remdesivir improved mortality if the duration from onset to treatment initiation was less than nine days [11]. The therapeutic intervention for patients hospitalized in our institution (median [IQR], six days [4,7]) was earlier than that in the previous reports. Almost all patients treated with remdesivir were discharged without sequelae from our hospital. These findings suggest that early treatment may improve clinical outcomes even in patients with non-severe COVID-19.

It has been reported that the fourth pandemic wave has been influenced by a rapid spread of the highly infectious B.1.1.7. strain [3]. In our study, about 70% of all hospitalized patients in the fourth pandemic period presented with this variant. More than 30% of patients required therapeutic intervention in this period, and the B.1.17 variant was the strongest risk factor for requiring systemic treatment. The treatment-requirement and the observation groups did not significantly differ in terms of duration from onset to hospitalization. Furthermore, the median duration was slightly longer in the treated group than in the observation group. In the Wakayama Prefecture, it was possible to accommodate all patients in the designated hospitals, and we believe that this strategy contributed to assessing the appropriate timing of therapeutic intervention. Some reports have shown that more contagious COVID-19 variants can occur and spread widely [13,14]. Treatment strategies for these variants should be further discussed.

There are numerous reports regarding risk factors among critically ill patients with COVID-19. In Japan, older age, underlying comorbidity, obesity, and smoking are considered risk factors of progression to severe illness [4]. Furthermore, our analysis showed that the need for therapeutic intervention during the fourth pandemic period was significantly higher in cases with underlying comorbidities and any systemic symptom—but not older age, obesity, and smoking. Moreover, serological biomarkers have been investigated, and results showed that C-reactive protein and interleukin-6 are associated with disease severity [15,16]. However, age and vital signs—not the aforementioned serological biomarkers—have been considered promising predictors [17]. In our hospital, patients with mild COVID-19 without risk factors did not undergo routine laboratory tests, and the current study could not assess serological biomarker levels—regardless, in many of the hospitals or clinics with less abundant resources, urgency and treatment options should be rapidly evaluated without detailed examination. Therefore, we focused on changes in vital signs among hospitalized patients with general risk factors including underlying comorbidities. The timing of therapeutic intervention in cases with rapid oxygen desaturation should be considered.

Fever, cough, and shortness of breath are the most common symptoms in patients with COVID-19 [18]. We also found that lower respiratory symptoms and a temperature ≥37.5 °C at admission were common among these hospitalized patients and were independent risk factors to require remdesivir treatment. Additionally, neurological symptoms are well known to be olfactory and taste disturbances. Headache occurs in around 10% to 20% of patients with COVID-19, which is less common than other symptoms [18,19]. Similar to these previous reports, our study also showed that headache was present in 15.7% of all hospitalized patients in the fourth pandemic. In the first half of 2020, two studies revealed that the clinical outcome was significantly improved in patients with COVID-19 with headaches [20,21]. On the other hand, many patients who required intubation were reported to present with headaches [22]. In the present study, patients hospitalized with COVID-19 who required therapeutic intervention were significantly more likely to have headaches. COVID-19 can potentially damage the central nervous system at the early phase and can paralyze respiratory sensors [23]. These findings suggest that patients with COVID-19 with headaches should also receive attention regarding the therapeutic intervention of remdesivir, in addition to the following factors: older age, diabetes mellitus, the B.1.1.7 variant, lower respiratory symptoms, and fever ≥ 37.5 °C.

There are a few limitations in the current study. First, the study used a small sample size and was conducted as a single-center retrospective study. Second, no blood testing was performed in all hospitalized cases. As described above, however, the importance of non-invasive and rapid evaluation should be underscored in resource-limited settings such as in pandemics. Third, the clinical severity of COVID-19 cases in areas surrounding Kihoku Hospital may be less severe compared to other areas.

## 5. Conclusions

Many patients with mild to moderate COVID-19 required remdesivir treatment during the fourth pandemic period. Moreover, risk factors, including ≥60 years of age, the B.1.1.7 variant, diabetes mellitus, headache, lower respiratory symptoms, and fever ≥37.5 °C, were found to be significant. Moreover, the hospitalization strategy based on the “Wakayama model” in Japan allowed us to carefully manage patients with mild to moderate COVID-19 presenting these risk factors through treatment with remdesivir.

## Figures and Tables

**Table 1 medicina-58-00118-t001:** Clinical characteristics and outcomes of COVID-19-infected patients in the fourth pandemic period according to treatment intervention.

	All *N* = 185	Treatment *N* = 63	Observation *N* = 122
Male, *n* (%)	94 (50.8%)	36 (57.1%)	58 (47.5%)
Age (years), mean (SD)	49.3 (19.8)	59.6 (14.7)	44.1 (20.1)
Comorbidity, *n* (%) Hypertension, *n* (%) Diabetes mellitus, *n* (%) Hyperlipidemia, *n* (%) Chronic obstructive pulmonary disease, *n* (%) Heart disease, *n* (%) Chronic kidney disease, *n* (%) Cancer, *n* (%)	81 (43.8%) 46 (24.9%) 17 (9.2%) 11 (5.9%) 5 (2.7%) 8 (4.3%) 3 (1.6%) 4 (2.2%)	35 (55.6%) 30 (47.6%) 14 (22.2%) 7 (11.1%) 5 (7.9%) 2 (3.2%) 3 (4.8%) 4 (6.3%)	25 (16.4%) 16 (13.1%) 3 (2.5%) 4 (3.3%) 0 (0%) 6 (4.9%) 0 (0%) 0 (0%)
Smoking, *n* (%)	50 (27.0%)	23 (36.5%)	27 (22.1%)
Body mass index (kg/m^2^), median (IQR)	23.1 (20.3–25.9)	23.9 (22.2–26.5)	21.6 (20–25.5)
COVID-19 B.1.1.7 variant, *n* (%)	125 (67.6%)	49 (77.8%)	76 (62.3%)
Asymptomatic at admission (excluding fever) Symptomatic at admission (excluding fever) Upper respiratory symptoms (nasal discharge, throat pain), *n* (%) Lower respiratory symptoms (cough, sputum), *n* (%) General fatigue, *n* (%) Headache, *n* (%) Arthralgia, *n* (%) Olfactory and taste disturbances, *n* (%) Gastrointestinal symptoms Shortness of breath, *n* (%)	64 (34.6%) 120 (65.4%) 54 (29.2%) 63 (34.1%) 32 (17.3%) 29 (15.7%) 11 (5.9%) 17 (9.2%) 14 (7.6%) 4 (2.2%)	12 (19.0%) 51 (81.0%) 31 (49.2%) 39 (61.9%) 23 (36.5%) 19 (30.2%) 11 (17.5%) 3 (4.8%) 8 (12.7%) 4 (6.3%)	53 (43.4%) 69 (56.6%) 23 (18.9%) 24 (19.7%) 9 (7.4%) 10 (8.2%) 0 (0%) 14 (11.5%) 6 (4.9%) 0 (0%)
Fever (°C) at admission, mean (SD)	37.2 (0.89)	38.0 (0.84)	36.8 (0.60)
SpO_2_ (%) at admission, median (IQR)	97 (95–98)	95 (94–95)	98 (97–98)
Pulse (/min) at admission, mean (SD)	83 (12)	87 (13)	81 (11)
Laboratory data at treatment initiation White blood cell count (/μL), median (IQR) Platelet count (×10^4^/μL), median (IQR) Alanine Aminotransferase (U/L), median (IQR) Aspartate Aminotransferase (U/L), median (IQR) Creatinine (mg/dL), median (IQR) Lactate Dehydrogenase (U/L), median (IQR) C-reactive Protein (mg/dL), median (IQR)	-	4640 (3230–5630) 14.8 (12.6–19.7) 26 (23–42) 31 (16–40) 0.88 (0.69–1.03) 234 (209–320) 2.52 (0.82–5.39)	-
Additional treatment information Oral baricitinib, *n* (%) Subcutaneous injection of heparin calcium, *n* (%)	-	12 (19.0%) 57 (90.5%)	-
* Age upon admission, mean (SD)	50.7 (19.8)	59.6 (14.7)	45.3 (20.6)
* Duration from onset to admission, median (IQR)	2 (1–4)	2 (2–4)	2 (1–4)
* Duration from onset to treatment initiation, median (IQR)	-	6 (4–7)	-
* Length of hospitalization, median (IQR)	10 (8–12)	10 (9–12)	9 (8-10)
Transferred to another hospital, *n*	1	1	0

IQR; interquartile range, SD; standard deviations. * In this analysis, 21 patients were excluded due to differences in discharge criteria.

**Table 2 medicina-58-00118-t002:** Comparison of patients hospitalized with COVID-19 with treatment requirement and clinical observation in the fourth pandemic period.

		Treatment	(*N* = 63)	Observation	(*N* = 122)	*p*-Value
Sex						0.277
	Male	36	(57.1%)	58	(47.5%)	
	Female	27	(42.9%)	64	(52.5%)	
Age (years)						<0.001
	<60	32	(50.8%)	94	(77.0%)	
	≥60	31	(49.2%)	28	(23.0%)	
Presence of comorbidity						
Hypertension						<0.001
	No	40	(63.5%)	105	(86.1%)	
	Yes	23	(36.5%)	17	(13.9%)	
Diabetes mellitus						<0.001
	No	49	(77.8%)	119	(97.5%)	
	Yes	14	(22.2%)	3	(4.1%)	
Hyperlipidemia						0.11
	No	56	(88.9%)	117	(95.9%)	
	Yes	7	(11.1%)	5	(4.1%)	
Chronic obstructive pulmonary disease						0.013
	No	59	(93.7%)	122	(100%)	
	Yes	4	(6.3%)	0	(0%)	
Heart disease						1
	No	61	(96.8%)	121	(99.2%)	
	Yes	2	(3.2%)	1	(0%)	
Chronic kidney disease						0.115
	No	60	(95.2%)	121	(99.2%)	
	Yes	3	(4.8%)	1	(0%)	
Cancer						0.038
	No	60	(95.2%)	122	(100%)	
	Yes	3	(4.8%)	0	(0%)	
Smoking history						0.054
	No	40	(63.5%)	95	(77.9%)	
	Yes	23	(36.5%)	27	(22.1%)	
Body mass index (kg/m^2^)						0.871
	<30	60	(95.2%)	114	(93%)	
	≥30	3	(4.8%)	8	(7%)	
COVID-19 genotype						0.031
	Previous	9	(14.3%)	39	(32.0%)	
	B.1.1.7	49	(77.8%)	76	(62.3%)	
	Unknown	5	(7.9%)	7	(5.7%)	
Symptoms at admission						
Upper respiratory symptoms						1
	No	46	(73.0%)	89	(73.0%)	
	Yes	17	(27.0%)	33	(27.0%)	
Lower respiratory symptoms						<0.001
	No	31	(49.2%)	92	(75.4%)	
	Yes	32	(50.8%)	30	(24.6%)	
General fatigue						<0.001
	No	40	(63.5%)	111	(91.0%)	
	Yes	23	(36.5%)	11	(9.0%)	
Headache						<0.001
	No	43	(68.3%)	111	(91.0%)	
	Yes	20	(31.7%)	11	(9.0%)	
Arthralgia						<0.001
	No	53	(84.1%)	122	(100%)	
	Yes	10	(15.9%)	0	(0%)	
Olfactory and taste disturbances						0.058
	No	61	(96.8%)	107	(87.7%)	
	Yes	2	(3.2%)	15	(12.3%)	
Gastrointestinal symptoms						<0.001
	No	51	(81.0%)	116	(95.1%)	
	Yes	12	(19.0%)	6	(4.9%)	
Shortness of breath						0.013
	No	59	(93.7%)	122	(100%)	
	Yes	4	(3.2%)	0	(0%)	
Fever at admission						<0.001
	<37.5 °C	30	(47.6%)	108	(88.5%)	
	≥37.5 °C	33	(52.4%)	14	(11.5%)	
SpO_2_ at admission						0.003
	≥97%	18	(28.6%)	64	(52.5%)	
	<97%	45	(71.4%)	58	(47.5%)	
Pulse at admission						<0.001
	<90/min	34	(54.0%)	97	(79.5%)	
	≥90/min	29	(46.0%)	25	(20.5%)	

**Table 3 medicina-58-00118-t003:** Univariate analysis of risk factors for therapeutic intervention of remdesivir in mild and moderate COVID-19 hospitalized patients.

	Odds Ratio (95% CI)	Univariate
Female	0.68 (0.37–1.25)	*p* = 0.22
≥60 years of age	3.25 (1.70–6.23)	*p* < 0.001
Presence of comorbidity		
Hypertension	3.55 (1.72–7.33)	*p* < 0.001
Diabetes mellitus	11.3 (3.12–41.2)	*p* < 0.001
Hyperlipidemia	2.92 (0.89–9.62)	*p* = 0.08
Chronic obstructive pulmonary disease	-	*p* = 0.99
Heart disease	0.77 (0.15–4.07)	*p* = 0.76
Chronic kidney disease	6.05 (0.62–59.4)	*p* = 0.12
Cancer	-	*p* = 0.99
Presence of smoking history	2.02 (1.04–3.94)	*p* = 0.039
Body mass index >30 kg/m^2^	0.71 (0.18–2.78)	*p* = 0.63
COVID-19 B.1.1.7 variant	2.79 (1.24–6.27)	*p* = 0.013
Symptoms at admission		
Upper respiratory symptoms	0.99 (0.50–1.98)	*p* = 0.92
Lower respiratory symptoms	3.17 (1.66–6.02)	*p* < 0.001
General fatigue	5.80 (2.60–13.0)	*p* < 0.001
Headache	3.86 (1.77–8.45)	*p* < 0.001
Arthralgia	-	*p* = 0.99
Olfactory and taste disturbances	0.23 (0.05–1.06)	*p* = 0.06
Gastrointestinal symptoms	4.55 (1.62–12.8)	*p* = 0.004
Shortness of breath	-	*p* = 0.99
Fever > 37.5 °C at admission	8.49 (4.03–17.9)	*p* < 0.001
SpO_2_ < 97% at admission	2.76 (1.44–5.29)	*p* = 0.002
Pulse > 90/min at admission	3.31 (1.71–6.42)	*p* < 0.001

95% CI; 95% confidence interval.

**Table 4 medicina-58-00118-t004:** Multivariate analysis with the adjusted odds ratio of risk factors for therapeutic intervention of remdesivir among mild to moderate COVID-19 hospitalized patients.

	Odds Ratio (95% CI)	Multivariate
≥60 years of age	6.90 (2.57–18.0)	*p* < 0.001
A previous history of diabetes mellitus	20.9 (3.11–140)	*p* = 0.002
COVID-19 B.1.1.7 variant	5.30 (1.65–17.0)	*p* = 0.005
Lower respiratory symptoms at admission	3.13 (1.30–7.53)	*p* = 0.011
Headache at admission	3.82 (1.37–10.6)	*p* = 0.011
Fever ≥ 37.5 °C at admission	4.55 (1.83–11.3)	*p* = 0.001

## Data Availability

All data in the present study are available from the corresponding author upon reasonable request.

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
