# Peer review of "Risk Factors for Therapeutic Intervention of Remdesivir in Mild to Moderate COVID-19—A Single-Center Retrospective Study of the COVID-19 Fourth Pandemic Period in Wakayama, Japan"

_medicina, 2022, doi:10.3390/medicina58010118_

Round 1

Reviewer 1 Report

1) Headache could be primary or secondary. It is not an alarming sign in the majority of the cases, except when there are interesting clinical findings associated with that. To a better comprehension, of headache as an alarming sign, it is needed more clinical characteristics about this sign to be described as important.

Do TP, Remmers A, Schytz HW, Schankin C, Nelson SE, Obermann M, Hansen JM, Sinclair AJ, Gantenbein AR, Schoonman GG. Red and orange flags for secondary headaches in clinical practice: SNNOOP10 list. Neurology. 2019 Jan 15;92(3):134-144. doi: 10.1212/WNL.0000000000006697. Epub 2018 Dec 26. PMID: 30587518; PMCID: PMC6340385.

2) Study participants. Only residents of Wakayama or also ‘‘immigrants’’ from other cities included in the study?

3) Clinical data. Medications in use by the subjects could not affect the results?

4) Clinical data. In the project of the study, the symptoms were obtained from open-ended or closed questions? If the symptoms are descriptive, this could change the statistical analysis. Please, if possible, provide the questionnaire model as supplementary material.

5) Clinical data. Describe how do you define the ‘‘treatment’’ and ‘‘observation’’ groups? Was it a planned strategy or were only they separated after the starting of the study?

6) All (mild, moderate, severe) cases, independently of O2 therapy, used Remdesivir?https://www.covid19treatmentguidelines.nih.gov/therapies/antiviral-therapy/remdesivir/

7) Statistical analysis.

a) Why were reputable software such R, SPSS, Stata not used? b) Why did you use three software for the analysis? c) How did you analyze if the distribution of your data? d) How the power of the study was calculated? e) Why was Fisher's exact test and not the chi-square test used? f) What are the variables analyzed in the logistic regression? g) What were the confounding variables? h) Were the logistic regression assumptions obeyed? i) Was there linearity? j) Was there multicollinearity?

https://webfocusinfocenter.informationbuilders.com/wfappent/TLs/TL_rstat/source/LinearRegression41.htm

8) Results. Could the author provide the spreadsheet of the cases as supplementary material?

Author Response

[Reviewer 1]

Thank you for your detailed reviewing on our manuscript. We are pleased that you presented very valuable comments and recommended publication with a minor revision. We have made every attempt to address your comments (please see point-by-point responses, below). Through addressing your comments, the manuscript is now much improved and acceptable to you for publication.

As you suggested, this manuscript has been proofread by a MDPI English Editing Services (Specialist) in this revision phase.

1) Headache could be primary or secondary. It is not an alarming sign in the majority of the cases, except when there are interesting clinical findings associated with that. To a better comprehension, of headache as an alarming sign, it is needed more clinical characteristics about this sign to be described as important.

Do TP, Remmers A, Schytz HW, Schankin C, Nelson SE, Obermann M, Hansen JM, Sinclair AJ, Gantenbein AR, Schoonman GG. Red and orange flags for secondary headaches in clinical practice: SNNOOP10 list. Neurology. 2019 Jan 15;92(3):134-144. doi: 10.1212/WNL.0000000000006697. Epub 2018 Dec 26. PMID: 30587518; PMCID: PMC6340385.

Response: Thank you for highlighting the important point. In this manuscript, the headache is among COVID-19-related neurological symptoms. These patients represented secondary headache caused by viral infection. In the section of 2.2. Clinical data, we have added the sentence as follows: “No patients have previous primary headache.”.

2) Study participants. Only residents of Wakayama or also ‘‘immigrants’’ from other cities included in the study?

Response: Thank you for your valuable comments. In this study, we included “only residents of Wakayama”, but not “immigrants”. So, we have added “residents of Wakayama” in the section of 2.1. Study participants.

3) Clinical data. Medications in use by the subjects could not affect the results?

Response: As pointed out, we did not refer the medications in use by the subjects. In the section of 2.1. Study participant, we have added the following sentence: “No enrolled patients had taken routine prophylactic administration of oral antipyretic or analgesic medications.”.

4) Clinical data. In the project of the study, the symptoms were obtained from open-ended or closed questions? If the symptoms are descriptive, this could change the statistical analysis. Please, if possible, provide the questionnaire model as supplementary material.

Response: Thank you for highlighting the important point. The symptoms were obtained from closed-ended questions. The questionnaire used in our hospital was written in Japanese, and an English translation has been attached as supplemental data. We have added the following sentence in the section of 2.2: “These symptoms were obtained from the questionnaire (supplemental data 1).”

5) Clinical data. Describe how do you define the ‘‘treatment’’ and ‘‘observation’’ groups? Was it a planned strategy or were only they separated after the starting of the study?

Response: We agree that the definition and the study design were confusable. In this study, we retrospectively reviewed medical records of 185 patients with COVID-19 who hospitalized in our hospital during the fourth pandemic period (from March 14 to May 31, 2021). Among them, 122 patients were carefully observed without treatment, whereas 63 patients required the treatment intervention. We defined the former as the "observation group" and the latter as the "treatment group. We have explained the definition in the first paragraph of the Results.

6) All (mild, moderate, severe) cases, independently of O2 therapy, used Remdesivir? https://www.covid19treatmentguidelines.nih.gov/therapies/antiviral-therapy/remdesivir/

Response: As suggested, this point had not been clearly stated. In our hospital, treatment strategy was based on the official Japanese guideline developed by the Japanese Ministry of Health, Labour and Welfare. In the section of 2.3. Treatment strategy in Kihoku Hospital, we have added these sentences as follows: “Briefly, patients with mild COVID-19 were hospitalized for careful monitoring. Remdesivir treatment was performed for COVID-19 patients with moderate I to II. Severe patients were transferred to other hospitals with intensive care units.”.

7) Statistical analysis.

  1. a) Why were reputable software such R, SPSS, Stata not used? b) Why did you use three software for the analysis? c) How did you analyze if the distribution of your data? d) How the power of the study was calculated? e) Why was Fisher's exact test and not the chi-square test used? f) What are the variables analyzed in the logistic regression? g) What were the confounding variables? h) Were the logistic regression assumptions obeyed? i) Was there linearity? j) Was there multicollinearity?

https://webfocusinfocenter.informationbuilders.com/wfappent/TLs/TL_rstat/source/LinearRegression41.htm

Response: As you pointed out, the explanation of statistical analysis was confusable and insufficient. Academic Editor also mentioned this point (comment 4).

a&b) We performed all analyses using the statistical software EZR. EZR is a graphical user interface for R (The R Foundation for Statistical Computing) based on R commander. We deleted two other software and explained the EZR in the section of 2.4. statistical analysis.

  1. c) The distribution of continuous variables was tested by the Shapiro-Wilk normality test. We have added this explanation in the section of 2.4. statistical analysis.
  2. d) Although data collection had already been completed in this retrospective study, we calculated the power for comparison between two proportions (a statistical significance level of α =0.05).
  3. e) In this study, we had conducted the chi-squared test between the two groups. Meanwhile, we used Fisher’s exact test if the sample size was small. We have amended this point in the section of 2.4. statistical analysis.
  4. f) The 23 variables analyzed in the univariate logistic regression include demographics, comorbidities, symptoms, and initial vital signs, that are associated with COVID-19 infection. Meanwhile, the variables that were significant in univariate logistic regression (p<0.01) were chosen as candidates for further multivariate analyses. We have added some sentences of these explanation in the section 3.4. and amended Table 4.
  5. g) Although we conducted propensity score estimation, confounding factors could not be confirmed among these variables. The C-index indicated the high score, 0.903 (exceeding 0.8).

h&i) To evaluate the assumption of linearity in our model, we checked likelihood ratio (LR) for the model fit statistics. LR test was statistically significant (p-value < 0.001) in the software used in this study. We suggest that there was the assumption of linearity in logistic regression. Moreover, regarding this analysis results, the curve (AUC) of the ROC curve by logistic regression was 0.903 [95%CI, 0.858-0.948].

  1. j) Because the variance inflation factor (VIF) for independent variables was less than 2.0, we suggest that there was no multicollinearity in this analysis.

8) Results. Could the author provide the spreadsheet of the cases as supplementary material?

Response: This study represented a small sample size and was conducted at a single-center retrospective study. The individual may be identified from the spreadsheet that includes age, gender, previous medical history, treatment strategy, and length of hospitalization etc. Therefore, it is difficult to provide the spreadsheet in handling the personal information.

Reviewer 2 Report

The authors present a retrospective electronic record based review of predetermined variables with the goal of determining likely outcomes from treatment.  they present the context and data clearly and because of the ongoing need for real time analysis of symptoms that could predict response to medications, this paper is of interest and will add some to our growing knowledge of neurologic symptoms and associations with Covid.  Simple but clear statistics are applied and suggest the importance of variables chosen and analyzed.  

Author Response

Thank you very much for your reviewing our manuscript. We are pleased that you found this manuscript to be well performed and acceptable for publication.

As you suggested, this manuscript has been proofread by a MDPI English Editing Services (Specialist) in this revision phase.

Reviewer 3 Report

Thank you for the well-written manuscript. It is clear and concise.

No comments and the only suggestion: you could maybe change the title to:

"Headache is an alarming sign for early treatment requirement in COVID-19 ..."

Author Response

[Reviewer 3]

Thank you for the well-written manuscript. It is clear and concise.

No comments and the only suggestion: you could maybe change the title to

"Headache is an alarming sign for early treatment requirement in COVID-19 ..."

Thank you for your detailed reading our manuscript. We are pleased that you found the article to be well performed, clearly presented and informative. We agree the valuable point you suggested. However, through addressing suggestions of the other Reviewer and the Academic Editor, we have amended this main text including the title. With these changes, we hope that the manuscript is now acceptable to you for publication.

As you suggested, this manuscript has been proofread by a MDPI English Editing Services (Specialist) in this revision phase.

Round 2

Reviewer 1 Report

1) OK 2) OK 3) OK 4) OK 5) OK 6) OK 7) OK 8) OK

-----------------------

CLARIFICATION/CORRECTION

i) Is right the number 140 in table 4? (A previous history of diabetes mellitus 20.9 (3.11–140) p = 0.002)

ii) Is right the number of smoking individuals in table 1? (Smoking, n (%) 50 (27.0%) 40 (63.5%) 95 (77.9%))

-----------------------

NEW IDEA/ SUGGESTION

A) Add ROC curve by logistic regression as a figure or supplementary material

B) Add a new figure relating the adjusted odds ratio of risk factors for therapeutic intervention of remdesivir among mild to moderate COVID-19 hospitalized patients. Like circles linking to these patients and other circles with the diameter according to the OR. It would be a great figure summary.

Author Response

[Reviewer 1]

Thank you for your detailed reading of our revised manuscript. We are pleased that you found the article to be well performed, clearly presented, and informative. During the revision phase, we have corrected the points you suggested. Through addressing your suggestions, this revised manuscript is now much improved. With these changes, we hope that the manuscript is now acceptable to you for publication.

As you suggested, this manuscript has been re-proofread by the Native speaker in this revised manuscript.

CLARIFICATION/CORRECTION

  1. i) Is right the number 140 in table 4? (A previous history of diabetes mellitus 20.9 (3.11–140) p = 0.002)

Response: We calculated this part again. This number is right.

  1. ii) Is right the number of smoking individuals in table 1? (Smoking, n (%) 50 (27.0%) 40 (63.5%) 95 (77.9%))

Response: Because this number was wrong, we have corrected this point.

-----------------------

NEW IDEA/ SUGGESTION

  1. A) Add ROC curve by logistic regression as a figure or supplementary material

Response: Thank you for your valuable comment. We have added this ROC curve as a figure (supplemental figure 2).”

  1. B) Add a new figure relating the adjusted odds ratio of risk factors for therapeutic intervention of remdesivir among mild to moderate COVID-19 hospitalized patients. Like circles linking to these patients and other circles with the diameter according to the OR. It would be a great figure summary.

Response: As suggested, we have added a new figure relating to the adjusted odds ratio of risk factors as supplemental figure 2.
